# Prolonged or Transition to Metabolically Unhealthy Status, Regardless of Obesity Status, Is Associated with Higher Risk of Cardiovascular Disease Incidence and Mortality in Koreans

**DOI:** 10.3390/nu14081644

**Published:** 2022-04-14

**Authors:** Juhee Lee, So-Young Kwak, Dahyun Park, Ga-Eun Kim, Clara Yongjoo Park, Min-Jeong Shin

**Affiliations:** 1Interdisciplinary Program in Precision Public Health, Graduate School, Korea University, Seoul 02841, Korea; jhlee1109@korea.ac.kr (J.L.); soyoung.kwak@nyulangone.org (S.-Y.K.); ekgus7171@korea.ac.kr (D.P.); gekim7@korea.ac.kr (G.-E.K.); 2Department of Population Health, NYU Grossman School of Medicine, New York, NY 10016, USA; 3Department of Food and Nutrition, Chonnam National University, Gwangju 61186, Korea; 4School of Biosystems and Biomedical Sciences, College of Health Science, Korea University, Seoul 02841, Korea

**Keywords:** metabolic health, obesity, cardiovascular disease, mortality

## Abstract

The risk of chronic disease and mortality may differ by metabolic health and obesity status and its transition. We investigated the risk of cardiovascular disease (CVD) and cancer incidence and mortality according to metabolic health and obesity status and their transition using the nationally representative Korea National Health and Nutrition Examination Survey (KNHANES) and the Ansan-Ansung (ASAS) cohort of the Korean Genome and Epidemiology Study. Participants that agreed to mortality linkage (*n* = 28,468 in KNHANES and *n* = 7530 adults in ASAS) were analyzed (mean follow-up: 8.2 and 17.4 years, respectively). Adults with no metabolic risk factors and BMI <25 or ≥25 kg/m^2^ were categorized as metabolically healthy non-obese (MHN) or metabolically healthy obese (MHO), respectively. Metabolically unhealthy non-obese (MUN) and metabolically unhealthy obese (MUO) adults had ≥1 metabolic risk factor and a BMI < or ≥25 kg/m^2^, respectively. In KNHANES participants, MUN, and MUO had higher risks for cardiovascular mortality, but not cancer mortality, compared with MHN adults. MHO had 47% and 35% lower risks of cancer mortality and all-cause mortality, respectively, compared to MHN. Similar results were observed in the ASAS participants. Compared to those persistently MHN, the risk of CVD was greater when continuously MUN or MUO. Transitioning from a metabolically healthy state to MUO also increased the risk of CVD. Few associations were found for cancer incidence. Using a nationally representative cohort and an 18-year follow-up cohort, we observed that the risk of CVD incidence and mortality and all-cause mortality, but not cancer incidence or mortality, increases with a continuous or a transition to an unhealthy metabolic status in Koreans.

## 1. Introduction

Cardiovascular disease (CVD) and cancer are the leading causes of death globally, accounting for 18.6 million and 10.0 million deaths, respectively, in recent years [1,2]. Obesity is positively associated with hypertension, diabetes, dyslipidemia, and increased risk of CVD, some cancers, and mortality [3,4,5]. However, a portion of obese adults has metabolically healthy phenotypes (normal blood pressure, glucose, and lipid levels). Whether these metabolically healthy obese (MHO) individuals are at higher risk of cardiovascular events, cancer, and mortality is not clear [6,7,8,9,10,11,12,13,14]. In addition, obesity and metabolic health status are not constant and can change over time [8,15,16]. Therefore, recent studies have additionally focused on the risk of CVD and cancer according to the transition of metabolic health and obesity status [17,18,19,20,21,22,23].

Most previous studies on metabolic health status and obesity were conducted on Whites. However, results in Whites may not be generalizable to Asian populations, due to the different body composition and fat distribution between Whites and Asians. Asians have more visceral fat and less subcutaneous fat with a similar body mass index (BMI) than Whites, which may explain the greater risk for morbidity and mortality in Asians at a lower BMI [24,25]. A few studies have investigated the risk of cardiovascular and cancer incidence and mortality according to metabolic health and obesity status and their change in Asians, however, they were with limited participants or based on medical expense claim data, which does not always correspond to medical diagnosis [26,27], and/or a relatively short follow-up (≤10 years) [18,21,23,28,29,30,31,32,33,34,35,36,37].

Recently, mortality data were linked to the Korea National Health and Nutrition Examination Survey (KNHANES). Using the nationally representative KNHANES years 2007–2015 and the Ansan–Ansung (ASAS) cohort of the Korean Genome and Epidemiology Study (KoGES), we assessed the relationship between metabolic health and obesity status on cardiovascular, cancer, and all-cause mortality risks. Additionally, the risk of CVD and cancer incidence according to the transition of metabolic health and obesity status was assessed in the ASAS cohort.

## 2. Materials and Methods

### 2.1. Data Source

This study was conducted using two datasets—KNHANES and the ASAS cohort of KoGES—which are both conducted by the Korea Disease Control and Prevention Agency (KDCA), targeting the free-living Korean population. The nationally representative KNHANES dataset was used to investigate the relationship between the combination of obesity and metabolic health status with cardiovascular, cancer, and all-cause mortality at the national level. The KoGES dataset was used to replicate the findings of KNHANES and additionally assess the association of transition to obesity and metabolic health status with CVD and cancer incidence and cardiovascular, cancer, and all-cause mortality. The detailed design and procedures of these cohorts are described elsewhere [38,39]. Briefly, KNHANES is a nationally representative cross-sectional study, composed of health examinations, health interviews, and nutrition surveys conducted in 1998, 2001, 2005, and annually since 2007. Data analyzed for the current study was obtained from the years 2007–2015. The ASAS cohort of KoGES is a community-based cohort study, initiated in 2001–2002 in adults aged 40–69 years residing in Ansung and Ansan with biennial follow-up. The current study utilized data collected at baseline (2001–2002) up to the eighth follow-up survey (2017–2018). KNHANES and KoGES data were linked to death certificates and medical records from 1 January 2007 and 1 January 2001, respectively, to 31 December 2019, provided by Statistics Korea. All KNHANES and KoGES participants provided written consent. The current study protocol was approved by the Institutional Review Board at Korea University (KUIRB-2020-0291-01).

### 2.2. Study Population

Among the 73,353 adult participants of KNHANES 2007–2015, 51,575 participants (70.3%) agreed to mortality follow-up. We additionally excluded 23,106 participants for the following reasons: aged under 40 years (*n* = 15,875), currently pregnant (*n* = 4), missing information on BMI or metabolic health status (*n* = 3839), history of cancer or CVD (*n* = 3291), or died within the first year of follow-up (*n* = 98). The final number of eligible study participants was 28,468 (12,249 males and 16,219 females). Among the 10,030 participants of KoGES at baseline in 2001–2002, 7894 participants (78.7%) agreed to a mortality follow-up. We excluded pregnant females (*n* = 38), participants whose BMI or metabolic health status was missing (*n* = 35) and those with a history of CVD or cancer (*n* = 291) at baseline. None of the participants died within the first year of follow-up. Accordingly, 7530 participants were included in the analysis for a mortality follow-up. The flow charts of study participants for each study are shown in Figure 1.

### 2.3. Assessment of Metabolic Health Status and Obesity

Metabolic health status was defined by the presence of any of the three metabolic health risks: diabetes mellitus, hypertension, or dyslipidemia [28]. Diabetes mellitus was defined as having at least one of the following: fasting glucose ≥ 126 mg/dL, self-reported use of anti-diabetic medication, or a self-reported physician diagnosis of diabetes mellitus. The presence of hypertension was defined as a systolic blood pressure of ≥140 mmHg, diastolic blood pressure of ≥90 mmHg, or a self-report of taking antihypertensive mediations or a medical diagnosis of hypertension. Dyslipidemia was defined as a total cholesterol of ≥240 mg/dL, self-reported antihyperlipidemic medication use, or a self-reported medical diagnosis of dyslipidemia. The participants were classified as metabolically healthy (MH) or metabolically unhealthy (MU) based on the number of metabolic health risks (0 vs. 1–3, respectively). BMI was calculated as body weight (kg) divided by the square of the height (m^2^). The participants were classified as non-obese (N) or obese (O) when the BMI was < or ≥25 kg/m^2^, respectively, according to the Asia-Pacific guidelines by the World Health Organization [40] and the guidelines by the Korean Society for the Study of Obesity [41]. Participants were then classified as metabolically healthy non-obese (MHN), metabolically unhealthy non-obese (MUN), metabolically healthy obese (MHO), or metabolically unhealthy obese (MUO). For KoGES participants, the combination of metabolic health and obesity status was defined at each visit. The transition of metabolic health-obesity phenotype was determined by comparing phenotypes at baseline and the fourth follow-up (2009–2010; *n* = 6665).

### 2.4. CVD and Cancer Incidence and Mortality Ascertainment

The biennial follow-up design of the KoGES cohort allows for the detection of a new onset of disease. The incidence of CVD was defined by a new self-report of at least one of the following in KoGES participants: physician diagnosis of CVD (i.e., myocardial infarction, congestive heart failure, coronary artery disease, cerebrovascular disease, or peripheral artery disease), use of stroke medication, or treatment for CVD. The new onset of cancer was also defined by a new self-report of at least one of the following in KoGES participants: physician diagnosis of cancer (i.e., lung cancer, gastric cancer, hepatocellular carcinoma, colorectal cancer, pancreatic cancer, uterine cancer, breast cancer, thyroid cancer, prostatic carcinoma, gallbladder cancer). Mortality data were ascertained by death records provided by Statistics Korea. The causes of death were classified according to the codes from the International Classification of Diseases, tenth version (ICD-10). We identified cardiovascular mortality (I00-I99; 778 deaths in KNHANES and 134 deaths in KoGES), cancer mortality (C00-D48; 1101 deaths in KNHANES and 227 deaths in KoGES), and all-cause mortality (3426 deaths in KNHANES and 624 deaths in KoGES). For sensitivity analyses, unnatural deaths due to external causes (V01-Y98; 330 deaths in KNHANES and 61 deaths in KoGES) were additionally excluded for all-cause mortality analyses.

### 2.5. Covariate Assessment

Demographic and behavioral data were obtained from questionnaires. Education level was categorized into four groups: elementary school graduate and below, middle school graduate, high school graduate, and university graduate and above. Household income was categorized into low (≤1,500,000 Korean won in KNHANES and <2,000,000 Korean won in KoGES) and high (>1,500,000 Korean won in KNHANES and ≥2,000,000 Korean won in KoGES). Smoking and drinking status were classified as current smokers/drinkers and non-smokers/drinkers. Current smokers/drinkers were defined as those who answered as smoking cigarettes or drinking alcoholic beverages at the time of the survey. The metabolic equivalents (METs), as a representative of physical activity, were calculated by summing the METs of each type of activity type (2.4 for light, 5.0 for moderate, and 7.5 for intense activities) multiplied by frequency/week.

### 2.6. Statistical Analysis

The characteristics of the four metabolic health and obesity status groups are presented as the mean (standard error of the mean) for continuous variables or as the number (percentage) for categorical variables and were compared by one-way analysis of variance or a chi-square test, respectively. For the assessment of mortality risk, participants were followed from survey entry to the date of death or the end of follow-up (31 December 2019), whichever occurred first. To investigate the risk of CVD and cancer incidence, KoGES participants were followed until the development of CVD or cancer or their last examination. The retention rates for KoGES were 85.8% (*n* = 8603), 74.9% (*n* = 7515), 66.7% (*n* = 6688), 66.5% (*n* = 6665), 62.2% (*n* = 6238), 58.9% (*n* = 5906), 63.0% (*n* = 6318), and 61.4% (*n* = 6157) at each follow-up survey, sequentially. All participants completed at least one follow-up. The Cox proportional hazard regression model was used to calculate hazard ratios (HRs) and 95% confidence intervals (CIs) for cancer incidence, cardiovascular events, cardiovascular and cancer mortality, and all-cause mortality according to metabolic health and obesity status. Additionally, the prevalence of each phenotypic group at each visit was examined in KoGES participants who completed all eight follow-up examinations (*n* = 3995). The association between the transition of phenotype from baseline to the fourth visit and cancer or cardiovascular event was investigated in participants with data for both visits (*n* = 6665). Cox models were adjusted for potential confounders: sex, age, residential area (urban or rural), education status, household income, current smoking status, current drinking status, and physical activity assessed as METs. For sensitivity analysis, we excluded underweight participants (BMI < 18.5 kg/m^2^), as the higher mortality risk of underweight adults compared to normal-weight adults may have confounded the association with mortality [42,43,44]. The complex survey design was taken into account for all analyses of KNHANES. All analyses were performed with Stata SE 13.0 (Stata Corp, College Station, TX, USA), and a two-tailed *p*-value < 0.05 was considered statistically significant.

## 3. Results

### 3.1. Baseline Characteristics of KNHANES Study Participants

Among the 28,468 participants in the nationally representative KNHANES study, 34.9% (*n* = 9944) of the participants were obese, and 54.0% (*n* = 15,377) were metabolically unhealthy (Table 1). The mean follow-up was 8.2 years. The proportions of MHN, MUN, MHO, and MUO were 36.4, 27.9, 12.7 and 23.0%, respectively. The mean ages of the metabolically healthy groups (MHN and MHO) were younger than those of the metabolically unhealthy groups (MUN and MUO). The mean BMI was approximately 22.3 kg/m^2^ in the non-obese groups and 27.4 kg/m^2^ in the obese groups. Among the metabolically unhealthy participants, the prevalence of each metabolic health component was lower in the non-obese group than in the obese group (23.4% vs. 26.4% for diabetes, 69.2% vs. 78.0% for hypertension, and 40.1% vs. 41.1% for dyslipidemia in the MUN and MUO groups, respectively, all *p* < 0.001). The mean number of metabolic health risks was higher in MUO compared to MUN adults (1.45 ± 0.01 vs. 1.33 ± 0.01, respectively, *p* < 0.001).

### 3.2. Risk of Cardiovascular, Cancer, and All-Cause Mortality According to Obesity and Metabolic Health Risk in KNHANES

Compared to MHN adults, metabolically unhealthy adults had an increased risk of cardiovascular mortality (HR: 1.77 [95% CI: 1.28–2.43] and 1.43 [95% CI: 1.01–2.01] for MUN and MUO, respectively; Table 2). In addition, MUN had a 19% higher risk [95% CI: 1.03–1.38] for all-cause mortality compared to MHN. On the other hand, MUN and MUO did not have an increased risk for cancer mortality. MHO adults had a comparable risk for cardiovascular mortality (HR: 0.81 [95% CI: 0.42–1.56]), a 47% lower risk for cancer mortality [95% CI: 0.34–0.82], and a 35% lower risk for all-cause mortality [95% CI: 0.49–0.87] compared to MHN. Results were similar when excluding underweight participants (data not shown). When obesity status was stratified as underweight, normal weight, and obese, metabolically unhealthy adults had a higher risk for cardiovascular mortality regardless of BMI (data not shown). In addition, the number of metabolic risks was positively associated with a risk for cardiovascular and all-cause mortality, regardless of obesity status (Appendix A). These results indicate that metabolic health may be a greater determinant of cardiovascular and all-cause mortality than obesity, but not of cancer. Obesity may rather be protective with respect to cancer mortality when metabolically healthy (HR: 0.53 [95% CI: 0.34–0.82]) for MHO compared to MHN). Results were similar when excluding deaths due to external causes (data not shown) and were similar between men and women when stratified by sex (Appendix A).

### 3.3. Baseline Characteristics of KoGES Study Participants

Currently, KNHANES does not have longitudinal follow-up data of participants, thus we are unable to measure the transition of metabolic health and obesity status or assess the incidence of disease. To investigate the effects of transition of metabolic health and obesity status and disease incidence using the community-based ASAS cohort of KoGES, we first replicated the results of KNHANES in the ASAS cohort. Participants (*n* = 7530) were followed for a mean of 17.4 years. Among the participants, 2937 (39.0%), 1356 (18.0%), 1579 (21.0%), and 1658 (22.0%) were classified as MHN, MUN, MHO, and MUO, respectively, at baseline (Table 3). Similar to the KNHANES participants, the metabolically healthy participants were younger than the metabolically unhealthy participants. The mean BMI was approximately 22.5 kg/m^2^ in the non-obese groups and 27.4 kg/m^2^ in the obese groups, which is comparable to KNHANES participants. The prevalence of diabetes mellitus, hypertension, and dyslipidemia was 19.5%, 73.8%, and 26.2% among subjects in MUN, and 18.8%, 79.0%, and 26.1% among MUO subjects, respectively. The mean number of metabolic health risks was higher in MUO compared to MUN adults (1.24 ± 0.01 vs. 1.20 ± 0.01, *p* < 0.001).

### 3.4. Risk of CVD and Cancer Incidence and Mortality and All-Cause Mortality According to Metabolic Health and Obesity Status at Baseline in the KoGES Cohort

The risk of mortality due to CVD, cancer, and all causes according to baseline metabolic health and obesity status in the KoGES cohort were similar to that of KNHANES participants (Table 4). Higher risks of cardiovascular incidence, cardiovascular mortality, and all-cause mortality were observed in metabolically unhealthy adults. On the other hand, the cardiovascular mortality risk of MHO was similar to that of MHN (HR: 1.01 [95% CI: 0.50–2.03]) despite the 36% higher risk of CVD incidence [95% CI: 1.07–1.73]. Risks of cancer incidence, cancer mortality, and all-cause mortality were lower in MHO compared to MHN, though they did not reach statistical significance (HRs: 0.88 [95% CI: 0.65–1.20], 0.77 [95% CI: 0.49–1.21], and 0.83 [95% CI: 0.62–1.12], respectively). Results were similar when external causes of death were excluded (data not shown).

### 3.5. Transition of Metabolic Health and Obesity Phenotypes and the Risk of CVD and Cancer

Participants with both baseline and the fourth KoGES follow-up data were included in the analyses of transition of metabolic and obesity status and disease incidence (*n* = 6665). These participants had similar characteristics to the above total KoGES mortality study population (data not shown). The prevalence of the four metabolic health-obesity phenotypes at each visit during the 16-year follow-up is presented in Figure 2. The prevalence of obesity was consistent at approximately 43% during the years. Notably, the proportion of MHN and MHO adults decreased by 57.3% and 49.1%, respectively, resulting in an increase in metabolically unhealthy adults (66.3% of the population by the eighth follow-up). The eight-year transition (from baseline to the 4th follow-up) of metabolic health and obesity phenotypes shows that the proportion of maintaining each metabolic health-obesity phenotype was relatively high in metabolically unhealthy participants (70.7% and 71.4% for MUN and MUO, respectively; Figure 3). However, only 45.9% of the MHO participants maintained this phenotype while 33.2% progressed to MUO.

The risk of CVD incidence was assessed according to the transition of metabolic and obesity phenotypes (Figure 4). Compared to individuals consistently MHN, those that persisted to be metabolically unhealthy were at more than a two-fold greater risk for CVD incidence (HR: 2.05 [95% CI: 1.56–2.70] and 2.39 [95% CI: 1.85–3.10] for MUN and MUO, respectively; Figure 4). Among adults that were MHN or MHO at baseline, those that transitioned to MUO by the fourth follow-up had a 1.90 [95% CI: 1.02–3.55] and 1.69 [95% CI: 1.10–2.59] fold greater risk for CVD incidence compared to participants that were steadily MHN or MHO, respectively. Similarly, a greater risk for CVD incidence was observed when the MHN or MHO participants transitioned to MUN (HR: 1.30 [95% CI: 0.93–1.82] and 1.69 [95% CI: 0.88–3.26], respectively) compared to their phenotypically stable counterparts, though were not statistically significant. On the other hand, the transition from MUN to MHN was associated with a 40% decreased risk for CVD incidence [95% CI: 0.36–0.996]. The results indicate that being metabolically unhealthy is associated with an increased risk of CVD events, but this risk may be prevented by improving metabolic health status. On the other hand, obesity did not increase the risk of CVD in metabolically healthy adults, as observed when comparing consistently MHN adults with consistently MHO adults (HR: 1.19 [95% CI: 0.81–1.75]) or with those that transitioned from MHN to MHO (HR: 1.68 [96% CI: 0.90–3.14]), or between adults consistently MHO and those that transitioned to MHN (HR: 1.12 [95% CI: 0.58–2.15]). However, among metabolically unhealthy adults, weight reduction decreased the risk of CVD incidence, as observed by the 37% decreased risk [95% CI: 0.41–0.96] in those that transitioned from MUO to MUN compared to adults who were steadily MUO.

Compared to adults with consistent MHN, stable MHO had a lower risk of cancer incidence, though the HR did not reach significance (HR: 0.76 [95% CI: 0.48–1.18]; Table 5). Few associations were observed between the transition of metabolic health and obesity status and cancer incidence. Compared to those with a stable MHO phenotype, adults that transitioned from MHO to MHN had a higher risk for cancer (HR: 1.95 [95% CI: 1.02–3.76]). These results indicate that MHO may be more protective of cancer incidence compared to MHN.

## 4. Discussion

The risks of CVD, cancer and all-cause mortality, according to metabolic health and obesity status, were examined in Koreans using two large databases—a nationally representative database and the ASAS cohort of KoGES. Additionally, we examined the risks of the incidence and mortality of CVD and cancer and the risk of all-cause mortality according to metabolic health and obesity status and its transition in Koreans using a biennial 18-year follow-up cohort. In both cohorts, individuals with metabolically unhealthy phenotypes exhibited a higher risk for cardiovascular and all-cause mortality than MHN individuals. Among metabolically healthy adults, transition to MUO increased the risk of CVD incidence. In metabolically unhealthy adults, changing to a healthy metabolic profile or non-obese weight decreased the risk of CVD incidence. On the other hand, MHO adults did not have an increased risk for cancer incidence, cardiovascular or cancer mortality, or all-cause mortality compared to MHN.

The relationships of baseline metabolic health and obesity status with cardiovascular, cancer, and all-cause mortality were similar in both KNHANES and KoGES cohorts, despite some discrepancies in baseline characteristics and length of follow-up. Compared to the nationally representative KNHANES participants, the KoGES ASAS cohort participants were physically active, more likely to live in rural areas, be less educated, have lower household income, and refrain from alcoholic beverages. Still, overall, the directions of the HR were very similar between the two cohorts. Our analyses of nationally representative data (KNHANES) with 8 years of follow-up were similar to that of KoGES (17 years of follow-up). The results were also in line with some previous studies despite the participants’ younger mean age [30] or shorter follow-up within 5 years [24,29] or 10 years [31,32] in other studies. The use of KNHANES ensures to capture a nationally representative sample of non-institutionalized adults. Institutionalized adults may have underlying health conditions, which may introduce bias to the present results. On the other hand, although previous large studies in Koreans using the National Health Insurance System database excluded patients previously diagnosed with CVD or cancer, they did not exclude hospitalized patients or adults with other severe medical issues [24,29]. In addition, these previous studies utilized medical expense claim data of which approximately 30% do not match the primary diagnostics codes, resulting in possible errors in identifying CVD and cancer incidence [27,28]. Still, similar results indicate that metabolically unhealthy Korean adults have a higher risk of mortality.

The present study shows that both long-term and short-term metabolic health and obesity are associated with cardiovascular morbidity and mortality, in addition to all-cause mortality. Compared to those constantly MHN, adults that maintained MUN or MUO phenotypes from baseline throughout the follow-up had higher risks of CVD incidence. However, the risk of CVD was weaker in adults that transitioned from MHN to a metabolically unhealthy state (MUN or MUO; Figure 4) indicating that a longer duration of a metabolically unhealthy state may worsen the risk of CVD. Still, relatively short-term metabolic health status cannot be ignored. Among MHO adults, those that transitioned to MUO had an increased risk of CVD compared to adults that were continuously MHO, in line with previous studies in Chinese with a shorter follow-up [18]. In addition, recovery from a metabolically unhealthy phenotype to a metabolically healthy phenotype decreased the risk of CVD incidence. This may explain why studies with relatively short follow-ups have similar results [18,24,29]. Our results show that a prolonged metabolically deleterious status may increase CVD incidence and mortality risk, but this risk may be attenuated if the duration is shortened. Therefore, continuous efforts to improve and maintain metabolic health are required to prevent cardiovascular and all-cause mortality.

We show that compared to adults MHN at baseline, those that were MHO have a higher risk of CVD incidence, but a lower risk of cardiovascular mortality, cancer incidence, and mortality, and all-cause mortality. Several previous studies, including those with Asians, have similarly reported that MHO individuals have an increased risk for incident CVD than MHN controls [10,11,24,32,33,34] but lower risk of cardiovascular mortality [24,29,30,31]. The inconsistency between the higher risk of CVD incidence and the lower risk of CVD mortality in MHO adults compared to MHN adults is not fully understood. One possible explanation is that as a high proportion of adults that were MHO at baseline transition to MUO, the risk of CVD also increases. However, due to the relatively short duration of being metabolically unhealthy, the risk of death is similar to MHN and not as high as those that were MUO from baseline. Another possibility is that cardiovascular events may be worsened by detrimental metabolic status resulting in fatal cardiovascular events in MUO adults, but not in MHO individuals. Regarding cancer, we found that the MHO phenotype may be protective of cancer incidence and mortality. These results are consistent with some previous reports in the US [34,45,46]. In Koreans, previous studies resulted in a higher risk of breast cancer or colorectal cancer incidence in MHO compared to MHN [35,36]. We did not observe a higher risk of cancer incidence, possibly owing to that we were unable to assess incidence by cancer type. In addition, breast and prostate cancers have a relatively high survival rate [47]. Therefore, the incidence of cancer may not have affected cancer mortality, reflecting the lower risk of total cancer mortality of MHO adults in our study and previous studies with a shorter follow-up [28,37]. With respect to all-cause mortality, the risk was similar to or higher in MHO compared to metabolically healthy and normal-weight Whites [31,48,49]. However, a meta-analysis in Asians resulted in a lower risk of all-cause mortality in MHO adults compared to MHN adults [28,50], similar to the results of the current study. The mechanism for the protective effect of obesity on cancer and mortality in metabolically healthy adults is not clear [12]. Our results, in addition to that of others [50], indicate that racial differences may exist, especially for all-cause mortality. For instance, in the US, Asian-Americans have a higher rate of diabetes mellitus compared to non-Hispanic Whites despite their lower BMI [51]. As diabetes mellitus is affected by genetic factors in addition to obesity and other lifestyle factors [52], the relatively low proportion of Asians that remain metabolically healthy despite obesity in their older years may also be genetically protected from cardiovascular, cancer, and all-cause mortality. More research on genetic or epigenetic effects is needed. Nevertheless, although obese, attaining metabolic health may prevent cardiovascular deaths, but not events, and cancer morbidity and mortality in Koreans.

Our study has some limitations. Although the follow-up period was as long as 18 years for KoGES participants, we did not have a sufficient number of deaths to calculate the effect of metabolic health and obesity status transition on mortality. In addition, due to the limited number of events, we were not able to assess the risk according to specific cardiovascular events or specific cancer types. As participants were community-dwelling adults, adults in nursing homes or other care facilities were not surveyed. Therefore, the results may not be generalizable to institutionalized adults. However, adults residing in care facilities are likely to have multiple health complications which may compromise the results. Obesity was assessed by BMI, not the proportion of body fat, a more precise measure of obesity. In addition, KNHANES and KoGES do not provide data on fat distribution, such as visceral fat. Despite these limitations, we believe that our study has its strengths. This is the first study to prospectively utilize the nationally representative KNHANES database, linked with mortality data, to assess the association between metabolic health and obesity status and mortality. In addition, the use of the follow-up data of the KoGES database enabled us to assess the transition of metabolic health and obesity status on CVD and cancer incidence. To reduce misclassification of obesity and metabolic health status, we adopted a strict MHO definition using anthropometric and biochemical measurements, as well as the use of medications, and self-reported doctor-diagnosis of each metabolic health risk. These may be more accurate than national health insurance claim data as used in some previous studies. The exclusion of those with CVD or cancer at baseline and the long follow-up suggests the possible causal relationship of metabolic health and obesity with CVD incidence and mortality. The use of the two databases, the long mean follow-up of KoGES (>17 years), and the consistent results strongly support the conclusions. Excluding deaths from external causes resulted in similar results, strengthening our confidence in the association between the risk of death and participants’ metabolic and obesity status.

## 5. Conclusions

In conclusion, we demonstrated through a nationally representative database and an 18-year biennial prospective cohort that poor metabolic health is associated with a higher risk of cardiovascular and all-cause mortality, while MHO is associated with a lower risk of cancer incidence and mortality and all-cause mortality. A persistent or a transition to unhealthy metabolic profiles increases the risk of cardiovascular incidence, emphasizing the importance of management of metabolic health.

## Figures and Tables

**Figure 1 nutrients-14-01644-f001:**
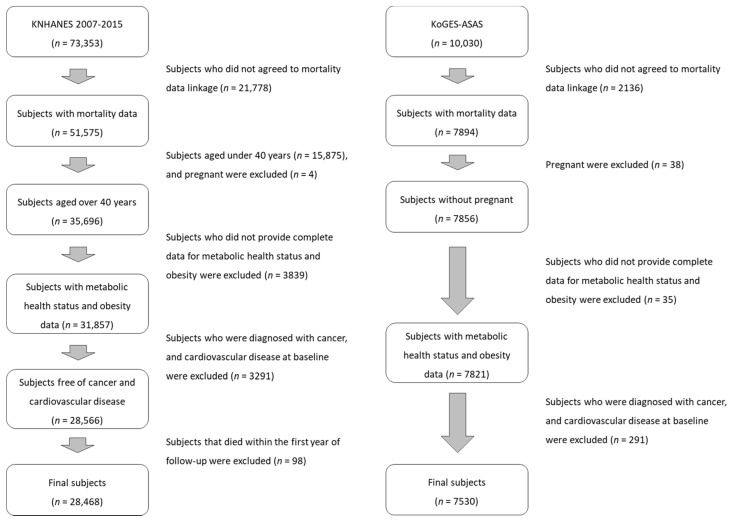
Flow chart of study population.

**Figure 2 nutrients-14-01644-f002:**
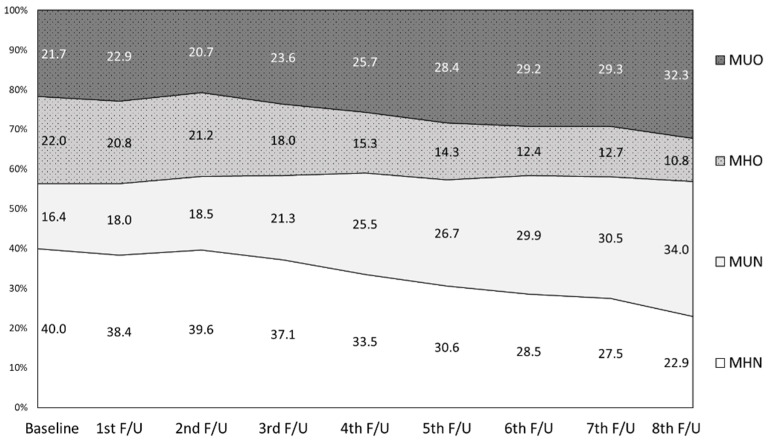
Transition of the combination of metabolic health risk and obesity during the follow-up period. Data were analyzed for those that had data for baseline and all 8 follow-up examinations (*n* = 3995). MHN: metabolically healthy non-obese; MHO: metabolically healthy and obese; MUN: metabolically unhealthy non-obese; MUO: metabolically unhealthy obese.

**Figure 3 nutrients-14-01644-f003:**
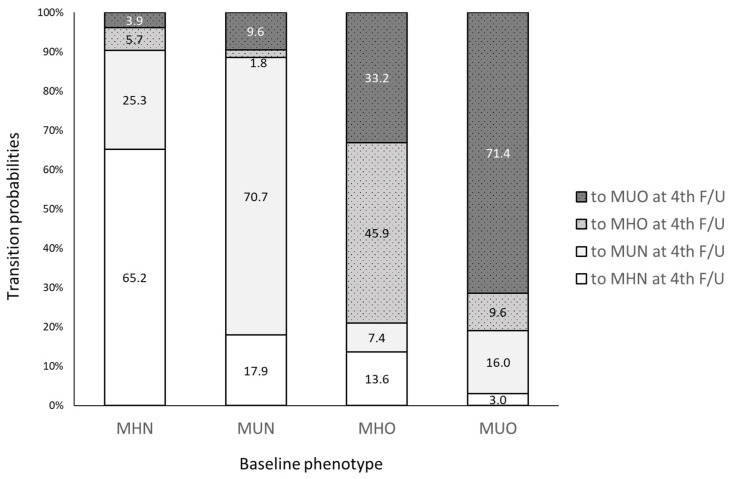
Transition of the combination of metabolic health risks and obesity status at baseline and fourth follow-up. MHN: metabolically healthy non-obese; MHO: metabolically healthy and obese; MUN: metabolically unhealthy non-obese; MUO: metabolically unhealthy obese (*n* = 6665).

**Figure 4 nutrients-14-01644-f004:**
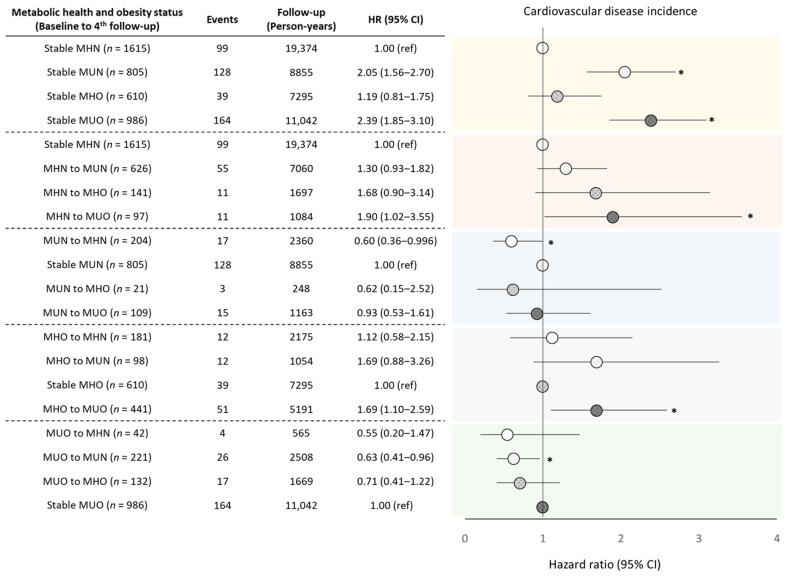
Hazard ratios (HRs) and 95% confidence intervals (Cis) for cardiovascular incidence according to the combination of metabolic health and obesity status at baseline and fourth follow-up. * *p*-value < 0.05. *p*-value was estimated using Cox proportional hazard regression model. MHN: metabolically healthy non-obese; MHO: metabolically healthy and obese; MUN: metabolically unhealthy non-obese; MUO: metabolically unhealthy obese; *n*: Number.

**Table 1 nutrients-14-01644-t001:** Characteristics of study population according to the combination of metabolic health risk and obesity status in KNHANES.

	MHN	MUN	MHO	MUO	*p*-Value
*n*	9813	8711	3278	6666	-
Weighted *n* (weighted %)	6,703,404 (36.4)	5,133,971 (27.9)	2,335,099 (12.7)	4,238,314 (23.0)	-
Lifestyle and socio-economic factors					
Age, years	51.2 ± 0.1	58.8 ± 0.2	50.6 ± 0.2	56.5 ± 0.2	<0.001
Male, *n* (%)	3900 (44.4)	3935 (50.0)	1435 (51.5)	2979 (52.5)	<0.001
Residential area, % Urban Rural	69.8 30.2	68.7 31.3	66.8 33.2	67.2 32.8	0.009
Education level, % ≤Elementary school graduate Middle school graduate High school graduate ≥University graduate	18.6 13.6 38.9 28.9	36.9 15.5 29.8 17.8	20.6 15.6 37.2 26.7	35.0 15.3 29.4 20.3	<0.001
Household income level, % Low (<₩2,000,000) High (≥₩2,000,000)	37.8 62.2	51.0 49.0	39.7 60.3	48.3 51.7	<0.001
Current smoker, *n* (%)	1894 (22.8)	1632 (22.8)	603 (23.1)	1110 (21.4)	0.325
Current drinker, *n* (%)	4938 (54.2)	4138 (53.6)	1704 (56.2)	3247 (55.4)	0.093
Metabolic equivalent of task	2235 ± 51	2040 ± 52	2296 ± 83	2070 ± 61	<0.001
Metabolic health risks and obesity status					
BMI, kg/m^2^	22.05 ± 0.02	22.56 ± 0.02	27.04 ± 0.04	27.54 ± 0.04	<0.001
Diabetes, *n* (%)	-	2037 (23.4)	-	1793 (26.4)	<0.001
Hypertension, *n* (%)	-	6219 (69.2)	-	5272 (78.0)	<0.001
Dyslipidemia, *n* (%)	-	3458 (40.1)	-	2760 (41.1)	<0.001
Number of metabolic health risks	-	1.33 ± 0.01	-	1.45 ± 0.01	<0.001

The analyses accounted for the complex sampling design (sampling weight, cluster, and strata) and the weighted mean values and percentages are presented. Values are presented as mean ± standard error for continuous variables or number of counts and percentage for categorical variables. *p*-values are from a one-way analysis of variance for continuous variables and the chi-square test for categorical variables assessing the difference according to the metabolic health risk and obesity status. BMI: body mass index; KNHANES: Korea National Health and Nutrition Examination Survey; MHN: metabolically healthy non-obese (BMI < 25 kg/m^2^ & 0 metabolic health risk); MHO: metabolically healthy and obese (BMI ≥ 25 kg/m^2^ & 0 metabolic health risk); MUN: metabolically unhealthy non-obese (BMI < 25 kg/m^2^ & 1–3 metabolic health risks); MUO: metabolically unhealthy obese (BMI ≥ 25 kg/m^2^ & 1–3 metabolic health risks); *n*: Number.

**Table 2 nutrients-14-01644-t002:** Hazard ratios (HRs) and 95% confidence intervals (CIs) for mortality according to the combination of metabolic health risks and obesity status in KNHANES.

	Weighted Total (*n*)	Weighted Events (*n*)	Weighted Follow-Up (PY)	Weighted Incidence Rate (per 1000 PY)	HR (95% CI)
Cardiovascular mortality					
MHN	6,703,404	34,649	53,880,020	0.64	1.00 (ref)
MUN	5,133,971	102,600	39,875,741	2.57	1.77 (1.28–2.43)
MHO	2,335,099	6503	18,997,511	0.34	0.81 (0.42–1.56)
MUO	4,238,314	45,969	33,735,295	1.36	1.43 (1.01–2.01)
Cancer mortality					
MHN	6,703,404	94,389	53,880,020	1.75	1.00 (ref)
MUN	5,133,971	119,614	39,875,741	3.00	0.91 (0.72–1.15)
MHO	2,335,099	13,718	18,997,511	0.72	0.53 (0.34–0.82)
MUO	4,238,314	77,409	33,735,295	2.29	0.93 (0.72–1.19)
All-cause mortality					
MHN	6,703,404	241,938	53,880,020	4.49	1.00 (ref)
MUN	5,133,971	427,964	39,875,741	10.73	1.19 (1.03–1.38)
MHO	2,335,099	40,672	18,997,511	2.14	0.65 (0.49–0.87)
MUO	4,238,314	202,375	33,735,295	6.00	0.92 (0.78–1.08)

The analyses accounted for the complex sampling design (sampling weight, cluster, and strata) and the weighted values are presented. The HRs (95% CI) were calculated using a Cox proportional hazard regression model adjusting for age, sex, residence area, education level, household income, smoking status, drinking status, and metabolic equivalent of task. BMI: body mass index; KNHANES: Korea National Health and Nutrition Examination Survey; MHN: metabolically healthy non-obese (BMI < 25 kg/m^2^ & 0 metabolic health risk); MHO: metabolically healthy and obese (BMI ≥ 25 kg/m^2^ & 0 metabolic health risk); MUN: metabolically unhealthy non-obese (BMI < 25 kg/m^2^ & 1–3 metabolic health risks); MUO: metabolically unhealthy obese (BMI ≥ 25 kg/m^2^ & 1–3 metabolic health risks); *n*: Number; PY: person-years.

**Table 3 nutrients-14-01644-t003:** Characteristics of study population according to the combination of metabolic health risk and obesity status in KoGES at baseline.

	MHN	MUN	MHO	MUO	*p*-Value
*n* (%)	2937 (39.0)	1356 (18.0)	1579 (21.0)	1658 (22.0)	-
Lifestyle and socio-economic factors					
Age, years	50.3 ± 0.2	54.9 ± 0.2	49.6 ± 0.2	53.6 ± 0.2	<0.001
Male, *n* (%)	1382 (47.1)	737 (54.4)	670 (42.4)	769 (46.4)	<0.001
Residential area, % Urban Rural	48.1 51.9	58.0 42.0	44.0 56.1	54.0 46.0	<0.001
Education level, % ≤Elementary school graduate Middle school graduate High school graduate ≥University graduate	27.0 24.2 36.1 12.7	38.5 21.4 25.6 14.5	27.6 24.9 33.7 13.8	40.0 20.7 25.3 14.0	<0.001
Household income level, % Low (<₩2,000,000) High (≥₩2,000,000)	61.5 38.5	69.0 31.0	58.3 41.7	66.0 34.0	<0.001
Current smoker, *n* (%)	767 (26.4)	374 (27.9)	345 (22.2)	335 (20.5)	<0.001
Current drinker, *n* (%)	1407 (48.3)	666 (49.6)	737 (47.2)	768 (46.7)	0.371
Metabolic equivalent of task	9749 ± 119	10456 ± 186	9301 ± 154	9848 ± 158	<0.001
Metabolic health risks and obesity status					
BMI, kg/m^2^	22.34 ± 0.03	22.79 ± 0.05	27.15 ± 0.05	27.65 ± 0.05	<0.001
Diabetes, *n* (%)	-	265 (19.5)	-	311 (18.8)	<0.001
Hypertension, *n* (%)	-	1001 (73.8)	-	1310 (79.0)	<0.001
Dyslipidemia, *n* (%)	-	355 (26.2)	-	432 (26.1)	<0.001
Number of metabolic health risks	-	1.20 ± 0.01	-	1.24 ± 0.01	<0.001

Values are presented as mean ± standard error for continuous variables or number of counts and percentage for categorical variables. *p*-values are from one-way analysis of variance for continuous variables and chi-square test for categorical variables assessing the difference according to the metabolic health risk and obesity status. BMI: body mass index; KoGES: Korean Genome and Epidemiology Study; MHN: metabolically healthy non-obese (BMI < 25 kg/m^2^ & 0 metabolic health risk); MHO: metabolically healthy and obese (BMI ≥ 25 kg/m^2^ & 0 metabolic health risk); MUN: metabolically unhealthy non-obese (BMI < 25 kg/m^2^ & 1–3 metabolic health risks); MUO: metabolically unhealthy obese (BMI ≥ 25 kg/m^2^ & 1–3 metabolic health risks); *n*: Number.

**Table 4 nutrients-14-01644-t004:** Hazard ratios (HRs) and 95% confidence intervals (Cis) for the cardiovascular disease incidence and mortalities according to the combination of metabolic health risks and obesity status in KoGES.

	Total (*n*)	Events (*n*)	Follow-Up (Person-Years)	Incidence Rate (per 1000 Person-Years)	HR (95% CI)
Cardiovascular disease incidence					
MHN	2937	183	30,729	5.96	1.00 (ref)
MUN	1356	172	13,390	12.85	1.69 (1.36–2.09)
MHO	1579	118	16,512	7.15	1.36 (1.07–1.73)
MUO	1658	224	16,710	13.41	1.94 (1.59–2.38)
Cardiovascular mortality					
MHN	2937	31	51,329	0.6	1.00 (ref)
MUN	1356	38	23,303	1.63	1.64 (0.99–2.70)
MHO	1579	11	27,814	0.4	1.01 (0.50–2.03)
MUO	1658	39	28,791	1.35	2.01 (1.23–3.29)
Cancer incidence					
MHN	2937	134	31,000	4.32	1.00 (ref)
MUN	1356	59	13,965	4.22	0.94 (0.69–1.29)
MHO	1579	63	16,867	3.74	0.88 (0.65–1.20)
MUO	1658	86	17,585	4.89	1.00 (0.75–1.33)
Cancer mortality					
MHN	2937	81	51,329	1.58	1.00 (ref)
MUN	1356	50	23,303	2.15	1.03 (0.71–1.49)
MHO	1579	28	27,814	1.01	0.77 (0.49–1.21)
MUO	1658	51	28,791	1.77	1.02 (0.71–1.48)
All-cause mortality					
MHN	2937	185	51,329	3.6	1.00 (ref)
MUN	1356	170	23,303	7.3	1.41 (1.13–1.75)
MHO	1579	63	27,814	2.27	0.83 (0.62–1.12)
MUO	1658	149	28,791	5.18	1.30 (1.04–1.62)

The HRs (95% CI) were calculated using a Cox proportional hazard regression model adjusting for age, sex, residence area, education level, household income, smoking status, drinking status, and the metabolic equivalent of the task. BMI: body mass index; KoGES: Korean Genome and Epidemiology Study; MHN: metabolically healthy non-obese (BMI < 25 kg/m^2^ & 0 metabolic health risk); MHO: metabolically healthy and obese (BMI ≥ 25 kg/m^2^ & 0 metabolic health risk); MUN: metabolically unhealthy non-obese (BMI < 25 kg/m^2^ & 1–3 metabolic health risks); MUO: metabolically unhealthy obese (BMI ≥ 25 kg/m^2^ & 1–3 metabolic health risks); *n*: Number.

**Table 5 nutrients-14-01644-t005:** Hazard ratios (HRs) and 95% confidence intervals (CIs) for cancer incidence according to the combination of metabolic health risks and obesity status at baseline and fourth follow-up.

Metabolic Health and Obesity Status		Total (*n*)	Events (*n*)	Follow-Up (Person-Years)	Incidence Rate (per 1000 Person-Years)	HR (95% CI)
Baseline	4th Follow-Up					
Participants with stable metabolic and obesity status						
MHN	MHN	1615	90	19,412	4.64	1.00 (ref)
MUN	MUN	805	41	9326	4.40	0.91 (0.62–1.34)
MHO	MHO	610	25	7392	3.38	0.76 (0.48–1.18)
MUO	MUO	986	59	11,795	5.00	0.92 (0.65–1.31)
Participants with changing metabolic and obesity status						
MHN	MHN	1615	90	19,412	4.64	1.00 (ref)
	MUN	626	27	7243	3.73	0.80 (0.52–1.23)
	MHO	141	10	1687	5.93	1.37 (0.71–2.64)
	MUO	97	1	1131	0.88	0.18 (0.02–1.27)
MUN	MHN	204	12	2361	5.08	1.16 (0.61–2.22)
	MUN	805	41	9326	4.40	1.00 (ref)
	MHO	21	0	256	0	-
	MUO	109	2	1256	1.59	0.34 (0.08–1.41)
MHO	MHN	181	14	2171	6.45	1.95 (1.02–3.76)
	MUN	98	6	1084	5.53	1.54 (0.63–3.78)
	MHO	610	25	7392	3.38	1.00 (ref)
	MUO	441	16	5421	2.95	0.80 (0.42–1.51)
MUO	MHN	42	3	563	5.33	1.42 (0.44–4.56)
	MUN	221	12	2571	4.67	1.00 (0.52–1.91)
	MHO	132	9	1709	5.27	1.07 (0.51–2.25)
	MUO	986	59	11,795	5.00	1.00 (ref)

The HRs (95% CI) were calculated using a Cox proportional hazard regression model adjusting for age, sex, residence area, education level, household income, smoking status, drinking status, and the metabolic equivalent of the task. BMI: body mass index; MHN: metabolically healthy non-obese (BMI < 25 kg/m^2^ & 0 metabolic health risk); MHO: metabolically healthy and obese (BMI ≥ 25 kg/m^2^ & 0 metabolic health risk); MUN: metabolically unhealthy non-obese (BMI < 25 kg/m^2^ & 1–3 metabolic health risks); MUO: metabolically unhealthy obese (BMI ≥ 25 kg/m^2^ & 1–3 metabolic health risks); *n*: Number.

## Data Availability

Restrictions apply to the availability of some or all data generated or analyzed during this study to preserve patient confidentiality or because they were used under license. The corresponding author will, on request, detail the restrictions and any conditions under which access to some data may be provided. Data was obtained from Korea Centers for Disease Control and Prevention and are available with the permission of Korea Centers for Disease Control and Prevention. Data in this study were from the Korea National Health and Nutrition Examination Survey, Korea Disease Control and Prevention Agency and Cause of Death Statistics, Statistics Korea.

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
