# Peer review of "Prolonged or Transition to Metabolically Unhealthy Status, Regardless of Obesity Status, Is Associated with Higher Risk of Cardiovascular Disease Incidence and Mortality in Koreans"

_nutrients, 2022, doi:10.3390/nu14081644_

Round 1

Reviewer 1 Report

  1. It is necessary to make corrections in the English of the article.
  2. In the introduction, the authors stated that the BMI of Asian individuals does not reflect visceral fat tissue. However, the authors gave no data on the patients' visceral fat tissue. In the limitation section, a small explanation was made about this subject. The authors should elaborate a little on this situation.

Author Response

Answers for Reviewer’s comments

Ref. No.: nutrients-1667449

Title: Prolonged or transition to metabolically unhealthy status, regardless of obesity status, is associated with a higher risk of cardiovascular disease incidence and mortality in Koreans

The authors sincerely appreciate the time spent in reviewing this manuscript, and your advice to improve it. We revised the manuscript following your and reviewers’ queries and comments, and highlighted the corrected and revised parts of the text in red. Please, see the attached files (revised manuscript and point-by point answers to Reviewers’ comments). We hope that you find them satisfactory.

Reviewer 1

  1. It is necessary to make corrections in the English of the article.

Answer) Thank you for this comment. English has been corrected.

  1. In the introduction, the authors stated that the BMI of Asian individuals does not reflect visceral fat tissue. However, the authors gave no data on the patients' visceral fat tissue. In the limitation section, a small explanation was made about this subject. The authors should elaborate a little on this situation.

Answer) Following the reviewer’s comment, we revised the limitation section.

Reviewer 2 Report

Manuscript # 1667449

This manuscript tilledProlonged or transition to metabolically unhealthy status, regardless of obesity status, are associated with a higher risk of cardiovascular disease incidence and mortality” Authors have performed a cohort study and emphasized the role of metabolic health status independent of obesity in the regulation of cardiovascular disease incidence and related mortality. This is an interesting study for the Asian population to codetermine the risk factors for the development of cardiovascular disease and related mortality including cancer. I have the following comments for this study.

Positive comments:

  • This cohort study has used good numbers of population with a long follow-up duration (18 years).
  • The study is well designed with the exclusion of unhealthy and metabolically altered phenotypes with age group ≥ 40 years.
  • Authors have emphasized the management of metabolic health status to prevent CVD and related mortality, and poor metabolic health status is associated with increased risk of cardiovascular disease and all-cause mortality whereas metabolic healthy obese individual has a lower risk of cancer incident and all cause-mortality.   

Negative comments:

  • Although authors have emphasized the management of metabolic health status to control the CVD and related all-cause mortality. However, their study is limited to Asians and only the Korean populations. This could bring limitations to the study. Otherwise, the authors should think to change the title and include Asian and Korean population names in the title.
  • Authors should correct some typo error

Author Response

Answers for Reviewer’s comments

Ref. No.: nutrients-1667449

Title: Prolonged or transition to metabolically unhealthy status, regardless of obesity status, is associated with a higher risk of cardiovascular disease incidence and mortality in Koreans

The authors sincerely appreciate the time spent in reviewing this manuscript, and your advice to improve it. We revised the manuscript following your and reviewers’ queries and comments, and highlighted the corrected and revised parts of the text in red. Please, see the attached files (revised manuscript and point-by point answers to Reviewers’ comments). We hope that you find them satisfactory.

Reviewer 2

This manuscript tilled “Prolonged or transition to metabolically unhealthy status, regardless of obesity status, are associated with a higher risk of cardiovascular disease incidence and mortality” Authors have performed a cohort study and emphasized the role of metabolic health status independent of obesity in the regulation of cardiovascular disease incidence and related mortality. This is an interesting study for the Asian population to codetermine the risk factors for the development of cardiovascular disease and related mortality including cancer. I have the following comments for this study.

Positive comments:

This cohort study has used good numbers of population with a long follow-up duration (18 years).

The study is well designed with the exclusion of unhealthy and metabolically altered phenotypes with age group ≥ 40 years.

Authors have emphasized the management of metabolic health status to prevent CVD and related mortality, and poor metabolic health status is associated with increased risk of cardiovascular disease and all-cause mortality whereas metabolic healthy obese individual has a lower risk of cancer incident and all cause-mortality.  

Negative comments:

Although authors have emphasized the management of metabolic health status to control the CVD and related all-cause mortality. However, their study is limited to Asians and only the Korean populations. This could bring limitations to the study. Otherwise, the authors should think to change the title and include Asian and Korean population names in the title.

Answer) Thank you for raising this issue. We revised the title to ‘Prolonged or transition to metabolically unhealthy status, regardless of obesity status, is associated with higher risk of cardiovascular disease incidence and mortality in Koreans’.

Authors should correct some typo error

Answer) Thank you for this comment. We revised the typos in the manuscript.
